# DevOps-Gym: Benchmarking AI Agents in Software DevOps Cycle

**Yuheng Tang**[1],[*] **Kaijie Zhu**[1][*]**, Bonan Ruan**[2]**, Chuqi Zhang**[2]**, Michael Yang**[1]**, Hongwei Li**[1]**,
Suyue Guo**[1]**, Tianneng Shi**[3]**, Zekun Li**[1]**, Christopher Kruegel**[1]**, Giovanni Vigna**[1]**, Dawn Song**[3]**,
William Yang Wang**[1]**, Lun Wang**[4]**, Yangruibo Ding**[5]**, Zhenkai Liang**[2]**, Wenbo Guo**[1][†]

[1]UC Santa Barbara, [2]National University of Singapore, [3]UC Berkeley, [4]Google, [5]UC Los Angeles

## Abstract

Even though demonstrating extraordinary capabilities in code generation and software issue resolving, AI agents' capabilities in the full software DevOps cycle are still unknown. Different from pure code generation, handling the DevOps cycle in real-world software, including developing, deploying, and managing, requires analyzing large-scale projects, understanding dynamic program behaviors, leveraging domain-specific tools, and making sequential decisions. However, existing benchmarks focus on isolated problems and lack environments and tool interfaces for DevOps. We introduce DevOps-Gym, the *first end-to-end benchmark* for evaluating AI agents across core DevOps workflows: build and configuration, monitoring, issue resolving, and test generation, plus end-to-end pipeline tasks that require solving problems across all stages sequentially. DevOps-Gym includes 700+ real-world tasks collected from 30+ projects in Java and Go. We develop a semi-automated data collection mechanism with rigorous and non-trivial expert efforts in ensuring the task coverage and quality. Our evaluation of state-of-the-art models and agents reveals fundamental limitations: they struggle with issue resolving and test generation in Java and Go, and remain unable to handle new tasks such as monitoring and build and configuration. These results highlight the need for essential research in automating the full DevOps cycle with AI agents.

## 1 Introduction

Software DevOps refers to the end-to-end process of developing, delivering, deploying, and managing software projects. It is a critical yet labor-intensive process. With recent advances in LLMs and AI agents, the development part can be largely automated (e.g., LLMs can automatically generate code, resolve GitHub issues) (Xia et al., 2024; Zhang et al., 2024c; Yang et al., 2024a; Wang et al., 2025a; Li et al., 2025; Tang et al., 2025). However, the subsequent operational stages, such as system building, deployment, and monitoring, still demand substantial manual intervention.

Automating these complex operational tasks presents a distinct set of challenges that transcend traditional code generation. It requires the ability to analyze system runtime behavior, interact with domain-specific tools, and execute multi-step plans. For instance, diagnosing a memory leak necessitates a sequence of actions: repeatedly invoking monitoring tools (e.g., using `ps` to inspect process state, `iostat` to identify I/O bottlenecks) to track memory resource usage and interpreting the output to identify signals of potential anomalies. Such tasks, which depend on complex tool use, dynamic interaction with a live environment, and coherent decision-making, make them a compelling application domain for autonomous AI agents.

While numerous benchmarks exist for evaluating LLMs and AI agents, they predominantly focus on software development tasks such as code generation (e.g., HumanEval (Chen et al., 2021)), issue resolving (e.g., SWE-bench (Jimenez et al., 2024)), and test generation (e.g., SWT-bench (Mündler et al., 2024)). In contrast, work on the operational side remains limited and is often confined to simulated environments (Jha et al., 2025) or narrow infrastructure settings (Chen et al., 2025). Consequently, there exists a significant gap for a benchmark capable of evaluating the end-to-end DevOps

---

[*]Equal contribution. Contact: {`yuhengtang, kaijiezhu`}@ucsb.edu
[†]Correspondence to: Wenbo Guo <`henrygwb@ucsb.com`>.

capabilities of agents in realistic software projects. Furthermore, few benchmarks are explicitly designed for agentic systems, which must execute multi-step workflows that integrate code reasoning and generation with tool usage. Existing benchmarks also lack dynamic execution environments with tool-calling interfaces that enable realistic agent interaction and evaluation.

To address these gaps, we introduce **DEVOPS-GYM**, the first benchmark designed to evaluate AI agents across core DevOps workflows using real-world repositories and various DevOps tools. Unlike existing benchmarks that focus on single tasks in isolation with synthetic environment, DEVOPS-GYM provides: (1) Coverage of four essential DevOps stages that form the minimum viable pipeline: build and configuration for project deployment, monitoring for runtime problem detection, issue resolving for problem fixing, and test generation for patch validation; (2) Real-world tasks sourced from GitHub issues or synthesized tasks that mimic the complex real-world issue patterns; (3) Agentic workflows that require extensive tool use and multi-step planning; (4) A tool-augmented dynamic evaluation environment with standardized tool-calling interfaces and detailed metrics for different types of tasks.

We make several design choices and efforts to address the key technical challenges of constructing DEVOPS-GYM. First, we manually analyze a large set of real-world issues, particularly for monitoring, build and configuration stage, to categorize representative issue types and summarize failure patterns for crafting synthetic tasks. Second, we apply a rigorous filtering process to prevent data contamination. Third, we invest extensive expert engineering effort to reproduce tasks to ensure their correctness, which required reconstructing environments, dependencies, configurations, and specific inputs. This was a multi-round, time-consuming process. Even with coding agent assistance, it often exceeds 10 hours of expert work per task, especially with incomplete reports. Finally, we carefully designed our evaluation metrics to enable rigorous and scalable evaluation, and provided standardized tool interfaces in the terminal-bench format (TerminalBench, 2025). With these extensive efforts, we craft 704 tasks collected from 30+ real-world projects in Java and Go. Additionally, we create 14 end-to-end pipeline tasks that chain all four stages sequentially, testing agents' ability to maintain context across complete DevOps workflows.

Our evaluation of three widely used coding agents across 5 LLMs and 4 agentic frameworks shows that even state-of-the-art systems fall short. The top-performing agent achieves success rates of just 51.85% on build and configuration, 20.56% on monitoring, 23.87% on issue resolving, and 13.87% on test generation. To the best of our knowledge, DEVOPS-GYM is the first end-to-end DevOps benchmark featuring agent-specific tasks alongside a comprehensive evaluation platform, including environments, tool interfaces, and metrics. We will open-source DEVOPS-GYM, together with its evaluation framework and baseline implementations, and will continue to improve it to facilitate future research on AI agents for broader software engineering. The key findings are shown below.

- *Agents frequently fail at high-level planning, struggling to formulate correct sequences of actions for multi-step building and monitoring workflows.*

- *Agents consistently fail to use DevOps-specific tools correctly, particularly for building and monitoring. We hypothesize that these tools are out-of-distribution (OOD) for the base LLMs, which are seldom trained on relevant tool-use trajectories.*

- *Agents exhibit a limited ability to parse and reason about dynamic information, such as program states, runtime logs, and system status. They also struggle with long-context reasoning required for complex tasks such as monitoring.*

- *For issue resolving and test generation, although SOTA agents report strong performance on existing Python-based benchmarks, their performance drops significantly on our Java and Go tasks. We believe this may be due in part to data contamination. It also indicates that LLMs are not well-equipped to handle non-script languages (e.g., Java and Go), which involve compilation, more complex dependencies, and syntax.*

## 2 RELATED WORK

**Coding benchmarks.** Function-level coding benchmarks (Chen et al., 2021; Majd et al., 2019; Jain et al., 2024; Liu et al., 2023) are widely used as standard benchmarks for evaluating LLMs' code generation capabilities. Some recent efforts move toward repository-level evaluation, which

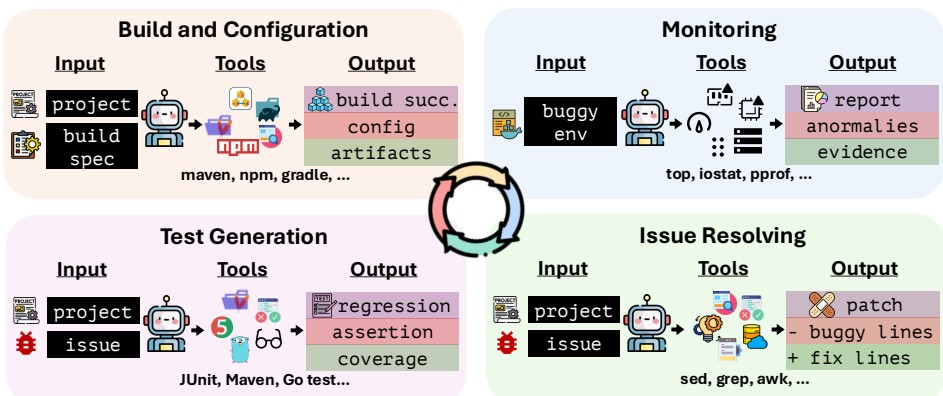

Figure 1: Overview of DEVOPS-GYM. It includes four core stages of DevOps: Build & Configuration, Monitoring, Issue Resolving, and Test Generation. Each stage requires an AI agent to leverage a distinct set of command-line tools to solve realistic tasks.

is more complex and relevant to real-world applications (Zhang et al., 2023; Zhuo et al., 2025; Ding et al., 2023; Li et al., 2024b; Liang et al., 2024; Han et al., 2025; Le Hai et al., 2024; Zhuo et al., 2024; Zhang et al., 2024a). For example, RepoCod (Liang et al., 2025) assesses whether LLMs can handle multi-file code generation rather than isolated functions. Another line of work extends general code generation to specific real-world tasks. For example, SWE-bench (Jimenez et al., 2024) targets issue resolving in real-world repositories for Python, and its follow-ups extend the effort to multilingual (Jimenez et al., 2024), multimodal (Zan et al., 2025; Ni et al., 2025), live settings (Jain et al., 2024), and security vulnerabilities (Yang et al., 2024b). Beyond issue resolving, task-specific benchmarks also include infrastructure (Kon et al., 2024; Munshi et al., 2025; Srivatsa et al., 2024), backend (Vero et al., 2025), or website and software development (Xu et al., 2025; Li et al., 2024a), database (Li et al., 2023), ML (Nathani et al., 2025) etc. Although offering more realistic evaluations, these benchmarks only cover the development side of DevOps instead of a holistic cycle. Furthermore, their main target is to evaluate model reasoning and generation capabilities rather than agentic systems with sequential decision-making and tool calls.

**DevOps-related benchmarks.** Beyond code development, there is an increasing effort in benchmarking other steps along the DevOps cycle, including test case generation (Zhang et al., 2024b; Wang et al., 2025b; 2024), build and configuration, and monitoring. Specifically, SWT-bench (Mündler et al., 2024) evaluates test case generation in Python, with metrics that measure functionality and code coverage. For build and configuration, existing benchmarks (Eliseeva et al., 2025) mainly focus on initial project configuration, evaluated by static and compilation checks for the repository. For monitoring, IT-bench (Jha et al., 2025) constructs tasks on monitoring and resolving issues during project operations. AIOpsLab (Chen et al., 2025) focuses on microservice environments and cloud-based incident detection, localization, and root cause analysis (e.g., network issues, authentication errors). Their evaluation metrics focus on the agent efficiency and costs such as the steps and time of actions, as well as the tokens consumed.

Limitations. There is no benchmark that covers the end-to-end DevOps cycle on real-world repositories. Most existing efforts focus on individual tasks (coding, testing, monitoring). Moreover, few benchmarks are explicitly designed for agentic systems, which have multi-step workflows that combine code reasoning and generation with tool callings. They also lack dynamic execution environments with tool calling interfaces for agent interaction and evaluation. To fill the gap, we construct **DEVOPS-GYM**, enabling the evaluation of AI agents on end-to-end real-world DevOps tasks with proper metrics, tool call support, and execution environments.

## 3 CONSTRUCTION OF DEVOPS-GYM

### 3.1 OVERVIEW

**Design principle.** We construct DEVOPS-GYM based on the following principles. ❶ Realism. We aimed to include real-world tasks that DevOps engineers encounter in practice. To achieve this, we

collect real-world GitHub projects and construct tasks based on actual GitHub issues or synthetic failures. The synthetic ones are injected by experts, which replicate common production problems, e.g., dependency conflicts from version incompatibilities. ❷ Agentic evaluation. We design our tasks to be sequential decision-making processes that involve various command-line tools. Solving these tasks requires domain-specific agentic systems capable of analyzing large projects, selecting proper tools and interpreting their returns, and conducting multi-step planning. For instance, to diagnose performance issues in a large project, such as abnormal memory or I/O utilization, agents must plan a multi-step procedure and invoke appropriate tools (e.g., `iostat`, `top`). ❸ Complete DevOps cycle coverage. We distill four critical stages: Build and configuration: configuring and migrating build systems, compiling the project, and fixing build failures; Monitoring: dynamically detecting runtime performance and resource issues during project executions; Issue resolving: debugging and patching bugs in projects; Test generation: creating regression tests that verify bug fixes.

**Benchmark overview.** We select Java and Go as our target program languages because they represent large-scale enterprise SE projects that have standardized, non-trivial build systems, as well as robust monitoring infrastructure and tooling. To ensure diverse coverage of realistic DevOps scenarios—particularly in monitoring and build, which are not covered in existing benchmarks—we manually analyze over $1,000$ GitHub issues from repositories cited in DevOps technical reports (DORA, 2024) and categorize the tasks for each stage. For build and configuration, we established two task categories: (i) repair tasks that rebuild the project to fix common failures in the general building process; and (ii) implementation tasks that build from scratch to incorporate new functionalities. For monitoring, we cover six anomalies: performance anomalies (I/O bottlenecks, query inefficiencies) and resource anomalies (memory leaks, disk exhaustion, CPU saturation, handle depletion). We follow Multi-SWE-bench (Zan et al., 2025) and SWT-bench (Mündler et al., 2024) to collect tasks for issue resolving and test generation. We further create end-to-end pipeline tasks that chain all four stages sequentially, requiring agents to maintain context from build repair through monitoring, issue resolution, and validation—testing their ability to handle complete DevOps workflows rather than isolated problems. Fig. 1 shows the pipeline of DEVOPS-GYM, comprising $54$ build and configuration tasks (20 synthetic tasks and 34 real-world tasks), $34$ monitoring tasks (29 synthetic tasks and 5 real-world tasks), 310 issue resolving tasks, 310 test generation tasks, and 18 end-to-end tasks, across $30+$ repositories. The process begins with building the project, where the agent must invoke build-related tools to either migrate an existing build or rebuild the project to incorporate new functionalities while resolving issues. After the build stage, the agent monitors the system's status and identifies runtime anomalies. When issues are detected, the agent is then responsible for resolving them and generating tests. Detailed benchmark statistics are in Appendix Table 5.

**Technical challenges and solutions.** Besides task selections and collection, which already require extensive expert efforts, constructing DEVOPS-GYM also encounters three key technical challenges. ❶ Data contamination prevention. Large-scale pre-training of LLMs poses significant contamination risks to new benchmarks. To address this, we applied a systematic prefix-completion analysis (Carlini et al., 2021; Staab et al., 2023) to identify and filter repositories potentially present in training corpora. Additionally, we sanitized the repositories by removing git metadata to prevent agents from accessing solutions through git version history (Kahn, 2025). Details are provided in Appendix B. ❷ Task reproduction. Reproducing real-world failures, especially for monitoring and building, imposes substantial challenges. We need to fully reconstruct the corresponding environments with correct run-time dependencies and configurations, as well as find the necessary inputs to trigger the issues. To make it even worse, the natural language issue descriptions typically do not provide the full information necessary for issue reproduction. Resolving this challenge also requires essential expert efforts to conduct extensive trial and error, i.e., it takes one expert SE researcher more than 10 hours to fully reproduce and validate one monitoring or building issue. Although coding agents (e.g., cursor and Claude code) can facilitate the process with hints, they still cannot fully finish the reproduction due to a lack of capabilities, as well as seeking shortcuts (we observe that instead of reproducing current errors, coding agents tend to inject other easier-to-trigger errors. Similar behavior has also been observed in other SE and security-related benchmarks (Wang et al., 2025c; Yang et al., 2024b)). ❸ Enable rigorous evaluation. After reproducing the issues, designing and implementing rigorous and scalable evaluation is also challenging, especially for building tasks. As detailed in Section 3.2, building tasks necessitate both dynamic execution validation and static configuration analysis. Besides, to provide a standardized interface for agent execution, we provide necessary tool sets (Figure 1) for each stage and convert our benchmark format into the terminal bench format (TerminalBench, 2025). In summary, the end-to-end task, selection, construction, and

evaluation pipeline requires extensive system design, engineering, and manual efforts that otherwise cannot be accomplished by SOTA automated agents.

## 3.2 TASK CONSTRUCTION FOR BUILD AND CONFIGURATION

**Overall task design.** Build and configuration is a critical step during the DevOps cycle, encompassing code compilation, dependency management, testing, and artifact creation within controlled environments. Also various errors can happen here including dependency conflicts, version incompatibilities, and configuration errors, frequently disrupting development workflows. To concretely understand agents' capability to perform complex build and configuration tasks, we evaluate the following two categories of build challenges that reflect routine DevOps scenarios. ❶ Repair tasks that address five prevalent project building error types: dependency version conflicts, build misconfiguration, compilation errors, tool-chain mismatches, and dependency resource unavailability. Agents must diagnose build failures by analyzing error logs, identifying root causes among these error categories, applying targeted fixes, and rebuilding the project to complete a correct build. ❷ Implementation tasks that incorporate new functionalities, which include the following scenarios: build system migration between frameworks (e.g., Maven to Gradle for Java), target release (i.e., release for specific use cases), plugin integration, and dependency version upgrades. Both categories require agents to understand build system semantics and configuration best practices essential to maintain reliable deployment pipelines.

For repair tasks, we follow the workflow of the BugSwarm framework for task collection (Tomassi et al., 2019), i.e., mining recent build failure-success pairs from CI logs and filtering for configuration-level fixes. For implementation tasks, three domain experts synthesize scenarios based on production environment patterns, ensuring coverage of frequent real-world build challenges.

**Key technical challenge and solutions.** Constructing build and configuration tasks present three key challenges that distinguish this domain from traditional code benchmarks. ❶ Issue reproduction: Real GitHub issues provide incomplete environment specifications, requiring careful reconstruction of tool-chain dependencies, compilation configurations, and version control to reproduce the issues. This challenging process requires extensive expert efforts; approximately 40% of initially selected issues required multiple iterations to achieve consistent reproducibility. ❷ Synthetic task design: Creating a comprehensive synthesis build task presents significant challenges. Implementing realistic build processes requires a deep understanding of repository configurations and compilation mechanisms, along with extensive DevOps experimentation to construct meaningful and challenging real-world scenarios. ❸ Enable accurate evaluation. Different from other steps, build requires a complex evaluation pipeline with multi-dimensional evaluation metrics. It requires designing different metrics for different task scenarios (detailed below). For example, for migration tasks, we need to design unified metrics that involve multiple configuration tools. Constructing an effective evaluation process for each selected task scenario also requires extensive engineering and manual effort for each individual task, e.g., designing unified metrics for multiple configurations requires manually analyzing the joint features with domain-specific tools.

**Task details.** The inputs, expected outputs, ground truth, and evaluation metrics are as follows:

- *Input:* (1) Repository with failing build configuration (repair tasks) or specification for new build setup (implementation tasks). (2) Terminal access with build tools (`maven`, `gradle`, `npm`), text utilities, and package managers.

- *Expected Output:* For repair tasks: patch (in the diff format) fixing build failure; for implementation tasks: complete configuration files meeting specifications.

- *Ground Truth:* For repair tasks: developer fixes from real repositories; For implementation tasks: expert-created configurations validated for correctness (implementation tasks).

- *Evaluation metrics:* At a high level, judging whether a build process is successful contains two sub-metrics: 1) the build process is executed without any errors; 2) the built artifacts correctly realize their required functionalities. For repair tasks, as the build command is standardized, as long as the agent executes the project's original build commands without errors, it can be considered as a successful build. For implementation tasks, different scenarios have different metrics for ensuring functionalities: migration between frameworks – whether the migrated implementation maintains functional equivalence with the original framework while successfully adapting

to the target framework's conventions and capabilities; target release – whether the released artifacts satisfied the required features; plugin integration – whether the plugin functions correctly; dependency version upgrades – successfully build the upgraded version; config initialization – successfully generate the configuration with all the functionalities we want. All these metrics are concretized as whether the built artifacts pass their dedicated testing cases.

### 3.3 TASK CONSTRUCTION FOR MONITORING

**Overall task design.** Given a running project application inside a controlled environment (e.g., a container), *monitoring* requires agents to (a) capture the runtime execution and underlying system states by using external command-line tools, and (b) detect potential performance and resource utilization anomalies during execution. Such a setting mirrors the role of real-world DevOps engineers, enabling agents to demonstrate their ability to diagnose realistic and complex production issues.

Importantly, our tasks focus on *performance and resource anomalies* rather than system failures or crashes. This is because immediate crashes can be simply identified through console outputs or error logs, limiting the opportunity to evaluate an agent's diagnostic capability. In contrast, performance and resource anomalies manifest as subtle system degradations that require careful analysis to uncover. Take memory leakage as an example. In a file-system server, developers introduce an in-memory cache for large file downloads (e.g., $\geq 2$ MB) but neglect to release it, which causes memory leakage when requesting various large files. While this leak eventually exhausts memory, the early symptoms appear as abnormal memory leak patterns relative to normal behavior (i.e., small file requests). Detecting such anomalies requires agents to *capture system behaviors across requests* and *analyze subtle variances between normal and buggy cases*.

Specifically, we consider two types of performance anomalies. ❶ Resource usage problems: Memory leaks, disk leaks, system handle (e.g., file and socket) exhaustion, and CPU spikes that gradually degrade system reliability. These issues represent the most common resource-related failures in production systems. Moreover, these problems exhibit gradual degradation patterns; for example, memory leaks may take hours or days to exhaust resources, and CPU spikes often manifest intermittently under specific load patterns. ❷ Performance degradations: We select I/O bottlenecks and inefficient SQL query handling that degrade user experience without causing immediate failures. For instance, a project mishandles I/O requests by opening files with `O_SYNC|O_DIRECT` without using proper OS-level caching or file-backend memory mappings. This would trigger extremely slow I/O paths when the request I/O payload size is large. Such degradations manifest as increased latency or reduced throughput, e.g., a $10\times$ slower request may still complete successfully. We also include cases where the system operates without any anomalies, requiring agents to correctly identify the absence of issues and avoid false positive diagnoses when monitoring healthy systems. Our tasks mix with real-world GitHub issues and expert-injected synthetic anomalies. In total, we collect 30 monitoring tasks, with the distribution of anomaly types shown in Fig. 3.

**Task-specific challenges and solutions.** Constructing monitoring tasks share similar challenges with build on ❶ Issue reproduction and ❷ Synthetic task design. Creating realistic expert-injected anomalies demands deep repository understanding to instrument source code without disrupting normal application behavior. We design anomalies that manifest within 5-15 minutes through standard monitoring tools while requiring sophisticated analytical reasoning rather than trivial detection. ❸ Observability validation: We manually validate that each task ensures anomalies while remaining detectable through monitoring toolsets (e.g, `top`, `free`, `ps`) without access to the source code.

**Task details.** The inputs, expected outputs, ground truth, and evaluation metrics are as follows:

- *Input:* (i) Containerized environment running an application with bugs; (ii) Terminal access (e.g., `top`, `free`, `ps`, `netstat`), with no access to source code, configuration files, or trigger scripts.

- *Expected Output:* Structured diagnostic report: specific issue (e.g., `memory_leak`), and supporting evidence with quantitative metrics (e.g., memory growth rate, affected process ID).

- *Ground Truth:* For real GitHub issues, we select only closed issues where the problem has been identified and resolved, and then the DevOps experts classify the issue based on the description and resolution. For expert-injected anomalies, ground truth is predetermined by the injection methodology as we know the exact problem type and manifestation because we control the failure

injection. To ensure reproducibility and observability, three senior DevOps engineers independently validate that each problem can be reliably detected using the provided monitoring tools.

- *Evaluation metrics:* The primary metric is binary accuracy, requiring agents to correctly identify the specific type of anomaly. In the prompt, we define the five anomaly types and explicitly instruct the model to write its diagnosis into a specified file in a single line without any explanation. Evaluation is performed using automated pytest scripts that check: (1) whether the diagnosis file exists, and (2) whether the diagnosed anomaly type matches the ground truth.

### 3.4 TASK CONSTRUCTION FOR ISSUE RESOLVING AND TEST GENERATION

**Overall task design.** These two stages represent well-established evaluation domains with existing benchmarks, such as SWE-bench (Jimenez et al., 2024) for issue resolving, as well as SWT-bench (Mündler et al., 2024) for test generation. Issue resolving requires agents to translate bug descriptions into code fixes, while test generation creates regression tests to prevent issue recurrence and ensure functionality correctness. Both tasks are essential components of the DevOps pipeline: fixes address problems identified through monitoring or user reports, and tests validate solutions before deployment. Following established methodologies, issue resolving agents receive buggy repositories with natural language descriptions and must generate patches that pass fail-to-pass test transitions (Jimenez et al., 2024). Test generation agents create tests based solely on bug descriptions to ensure comprehensive validation coverage.

**Key differences from existing benchmarks.** ❶ DEVOPS-GYM implements comprehensive decontamination procedures (Section 3.1) to mitigate potential training data contamination, ensuring more reliable evaluation than existing benchmarks. ❷ Cross-language performance gaps. As discussed in Table 1, LLM agents perform significantly worse on non-Python languages. This degradation likely stems from Python's dominance in training data.

**Data collection and evaluation.** We adapt the Multi-SWE-bench (Zan et al., 2025) collection pipeline for issue resolving tasks, targeting well-maintained Java and Go projects with comprehensive test suites. The pipeline filters GitHub pull requests that resolve issues with test-validated fixes, ensuring reproducible fail-to-pass transitions. Test generation tasks derive from the same issue set, creating corresponding validation scenarios. Regarding metrics, issue resolving success requires patches that pass all given test cases, while test generation success demands that generated tests fail on buggy code but pass the patched code. Task details can be found in Appendix C.

### 3.5 END-TO-END TASKS

Beyond evaluating individual DevOps capabilities, we construct end-to-end pipeline tasks that test agents' ability to maintain context and solve problems across multiple stages. These tasks simulate realistic DevOps workflows where problems cascade through the development pipeline.

Each pipeline task follows a four-stage sequence: **Stage 1: Build.** Starting with a monitoring-ready repository, we inject build configuration errors (e.g., missing dependencies, incorrect paths) that prevent compilation. Agents must diagnose and fix these issues to establish a working baseline. **Stage 2: Monitoring and Detection.** Once the build succeeds, agents deploy and monitor the running system, which contains a latent performance or resource issue from our monitoring task set (3.3). Agents must use system tools to identify anomalies without prior knowledge of the issue type. **Stage 3: Issue Resolving.** Based on monitoring findings, agents implement code-level fixes addressing the root cause. **Stage 4: Validation and Testing.** Agents rebuild the system with their fix and create regression tests that verify the issue is resolved. Tests must specifically target the performance characteristic identified in Stage 2, ensuring the problem cannot recur.

This pipeline design evaluates critical DevOps competencies absent from isolated task evaluation: maintaining problem context across tools, propagating diagnostic information between stages, and verifying end-to-end solution correctness. Success requires completing all four stages sequentially—failure at any point terminates the pipeline, reflecting real-world deployment constraints where partial solutions have no value.

Table 1: Evaluation results on DEVOPS-GYM for different agent frameworks and different LLMs. The best result for each stage are marked as **bold**.

| Agent | Model | Build & Config | Monitoring | Issue Resolving | Test Generation |
|---|---|---|---|---|---|
| OpenHands | Qwen3-Coder-30B | 20.37% | 5.89% | 13.22% | 6.13% |
| | o4-mini | 24.07% | 8.82% | 10.32% | 8.70% |
| | DeepSeek-V3.1 | 11.11% | 0.00% | 14.20% | 3.22% |
| | Gemini-2.5-Pro | 16.66% | 11.76% | 10.96% | 2.90% |
| | Claude-4-Sonnet | 42.59% | 14.70% | **23.87%** | 11.61% |
| mini-SWE-Agent | Claude-4-Sonnet | 29.62% | 2.91% | 5.16% | 0.98% |
| Aider | Claude-4-Sonnet | 5.55% | 0.00% | 9.67% | 2.25% |
| Claude Code | Claude-4-Sonnet | **51.85%** | **20.56%** | **23.87%** | **13.87%** |

## 4 EVALUATION

In this section, we conduct a comprehensive evaluation on DEVOPS-GYM using a suite of state-of-the-art agentic frameworks and reasoning models to identify their current strengths and weaknesses in automating the end-to-end DevOps lifecycle.

### 4.1 EXPERIMENTAL SETUP

**Agents and Models**. We evaluate three best-performing agentic frameworks: OpenHands (Wang et al., 2025a), mini-SWE-agent (Yang et al., 2024a), and Claude Code (Anthropic, 2025b). To comprehensively assess the impact of underlying backbone LLMs, we pair these frameworks with different LLMs. Specifically, we evaluated the OpenHands framework with five leading models: Claude-4-sonnet (Anthropic, 2025a), o4-mini (OpenAI, 2025), Gemini-2.5-Pro (Google, 2025), Deepseek-v3.1, and Qwen3-Coder-30B (Qwen-Team, 2025). This setup allows for a direct comparison of models within a single agent architecture. For the comparison of different agentic frameworks powered by the same underlying model, both mini-swe-agent and Claude Code are evaluated with Claude-4-sonnet as their backbone LLM. Our initial evaluation focuses on the most advanced agents and models to establish an upper bound on DEVOPS-GYM. We will continue to extend it to weaker models and agents to highlight the challenges in DevOps cycle automation.

### 4.2 RESULTS

**Cross-tool builds and configurations present new challenges to LLMs.** As a new yet complementary evaluation to existing benchmarks, DEVOPS-GYM provides the unique chance to evaluate agents' ability to reason about heterogeneous build ecosystems and switch across tools (e.g., Maven to Gradle) while repairing failures and implementing new functionalities, which remains largely under-evaluated. As shown in Table 1, model performance varies significantly during the build and configuration stage. Claude-4-Sonnet outperforms other state-of-the-art models by more than 20%, demonstrating its superior capabilities in large project configuration management and debugging. Regarding agent performance, even when using the same powerful Claude-4-Sonnet model, Claude-Code performs substantially better than the simpler mini-SWE-Agent. This performance gap indicates that complex build and configuration tasks require agents with sophisticated tool-call capabilities and well-designed architectures to manage challenging, multi-step processes effectively. Furthermore, all of the models perform poorly in the build implementation tasks, especially in migration tasks and target release tasks (see Appendix Table 6). From our observations, agents struggle with understanding the internal mechanisms of build tools like Maven and goreleaser (GoReleaser Team, 2024), as well as their practical usage patterns in real-world projects, rather than simply parsing error logs. This deeper knowledge gap becomes evident when agents attempt to configure or debug complex build processes. Detailed examples can be found in Appendix E.1. This is fundamentally different from fixing a bug in source code, where the context is more self-contained. This result demonstrates that while agents are improving at manipulating source code, they are far from capable of managing the software's build and deployment environment.

After examining the agent run logs, we identified three common error types for build and configuration tasks. First, toolchain and environment instrumentation limitations (33%). A common example

is unused-import violations. Similar trends appear in missing-dependency errors, malformed build files, and XML parsing failures, all of which reflect the agent's inability to validate or inspect configuration artifacts because the environment does not expose the necessary validators or schema checkers. These kinds of errors happen in the agent Openhands more than in Claude code. Second, multi-step reasoning and sequential planning failures (23%). In multi-step build repairs and build-system migrations, agents often resolve an initial error but lose track of remaining issues, revealing limitations in context retention. Many failures also occur because prompts do not enforce an iterative "fix-run-verify" loop, causing the agent to stop after addressing only part of the pipeline remaining failures. Third, domain-specific knowledge gaps (37%). Failures reflect genuine domain knowledge and deep technical understanding requirements that exceed current model capabilities. These include inherently complex tasks such as Maven-to-Gradle migrations that require understanding of build-system semantics, dependency resolution, or platform-specific constraints. Other failures arise from misunderstanding Java generics, type compatibility, JVM version differences, or subtle runtime behaviors such as NullPointerExceptions. These tasks represent the upper bound of DevOps complexity in our benchmark and quantify where agent abilities fundamentally fall short. Notably, 17% of all failures fall into the "inherently difficult" category, confirming that the benchmark captures real-world, high-difficulty DevOps scenarios.

**The dynamic nature of system monitoring reveals critical agent failures in processing continuous, temporal inputs.** As shown by Table 1, agents perform exceptionally poorly on monitoring tasks, even reporting 0% with the state-of-the-art models. This failure stems from the dynamic nature of monitoring, which poses three fundamental challenges to current LLM-based agents. First, monitoring requires the continuous processing of evolving system state information. Anomalies like memory leaks often manifest gradually over time, producing a long stream of observations. Agents must ingest this constant flow of new tokens, exhausting their context limit quickly even before a discernible issue appears. Second, we observe that agents struggle to consistently focus on monitoring, even though we prompt them to. Specifically, as we show in Appendix E.2, agents frequently become distracted, focusing excessively on analyzing earlier observations, which causes them to stop actively monitoring the live system state. This inability to balance long-term analysis with real-time awareness prevents them from detecting the subtle and sudden signals that define many real-world performance and resource issues. Third, agents exhibit poor baseline discrimination, generating significant false positives by misinterpreting normal operational variance as anomalies in healthy system environments. These monitoring failures reveal that current agents lack the temporal reasoning and sustained attention mechanisms essential for dynamic system observation.

**Unlike SWE-Bench, LLM agents perform poorly on DEVOPS-GYM for issue resolving**. As we can see from Table 1, the resolve rate drops significantly when moving from Python repositories, as we have seen in SWE-Bench (Jimenez et al., 2024), to other languages, such as Java and Go. This performance degradation is particularly striking when examining specific configurations: using the same agent and model combination (OpenHands + Claude-4-Sonnet), the resolving rate achieves 70.4% on the SWE-Bench-Verified leaderboard, yet drops dramatically to 23.87% when applied to Java and Go repositories in our benchmark. This indicates that the existing LLMs struggle with the cross-language capability gap, which might be largely due to the dominance of Python code in the training data. Also, results suggest that those complex compilation processes, dependency management, and environment build and configuration in Java and Go pose major challenges to Python-centric agents to overcome.

Though similar trends have also been reported by Multi-SWE-Bench(Zan et al., 2025), our evaluation provides a more definitive validation with a more thorough data decontamination (Section 3.1) provided by DEVOPS-GYM. Furthermore, by comparing our results using Claude-4 with those reported by Zan et al. (2025) using Claude 3.7, we confirm that, despite the significantly improved reasoning and tool-using capabilities of newer models, they still fail to overcome the major challenges in navigating the non-Python ecosystems.

**Generating high-quality tests for the issue is even more challenging than resolving the issue.** Interestingly, when using the exact same set of issues for evaluation, we find that the accuracy of generating high-quality tests is notably lower than the issue resolving rate. This highlights the general difficulty of test generation, which requires agents to not only have a static understanding of the whole repository but also the dynamic analysis capabilities that could reason about how the described bug would be triggered during execution. The agent also needs to reason about how the bug might be resolved, so that the generated tests could not only reproduce the described failure but also

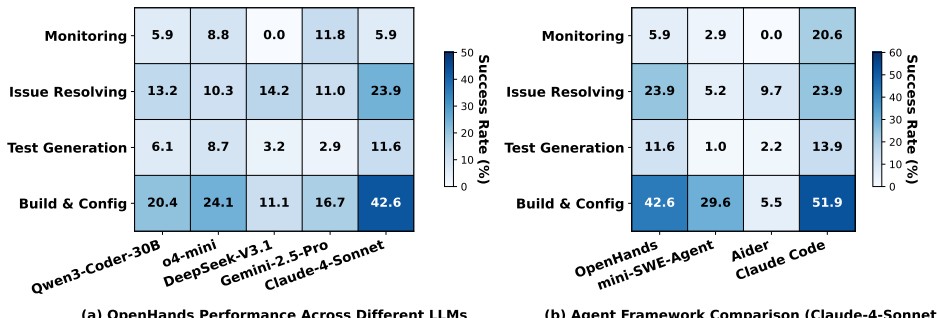

Figure 2: Performance comparison across different agentic frameworks and LLMs.

validate the correctness of the patch. In contrast, generating a patch can sometimes be accomplished through more straightforward, static code analysis when the issue description potentially discusses the expected fix. These results suggest that while agents are becoming proficient at predicting coding patterns, reasoning about runtime behavior remains a significantly more challenging goal to achieve.

Compared to SWT-Bench (Mündler et al., 2024), DEVOPS-GYM is more challenging for LLM agents to achieve satisfactory performance (as illustrated in Table 1) for two reasons. First, DEVOPS-GYM focuses on compiled languages (Java and Go), where reasoning about dynamic program behavior is inherently more complex than in interpreted languages like Python, which SWT-Bench evaluates, since understanding a multi-stage compilation and linking process adds obvious difficulties for agents. Second, while SWT-Bench incorporates code coverage as a soft metric for test effectiveness, we focus on the strict metric that a generated test must precisely reproduce the failure described in the issue and subsequently pass on the patched code.

**Analysis Across Different Agentic Frameworks and Models** Fig. 2 shows that Claude Code achieves the best overall performance, while Claude-4-Sonnet is the top-performing LLM across most tasks. Claude Code consistently outperforms other agentic frameworks, reaching 58.33% success in build and configuration. The substantial gaps between different agentic frameworks suggest that agent design is crucial for automated software engineering performance.

**End-to-end pipeline tasks, designed to test problem-solving across cascading DevOps stages, show the inability of current agents to handle long-horizon workflows.** Agents achieve **0%** success rate in completing all four stages on any single end-to-end task (detailed breakdown analysis are shown in Table 3). While agents can sometimes successfully resolve build configuration errors in Stage 1, they struggle to complete Stage 2 (Monitoring and Detection) due to the monitoring challenges discussed earlier—inability to process continuous temporal inputs and maintain sustained attention on system metrics. Interestingly, we observe that agents occasionally succeed in Stage 3 (Issue Resolving) or Stage 4 (Test Generation) even with incorrect monitoring conclusions, as different performance issues often share similar code-level manifestations that enable coincidental fixes. However, the overall pipeline success rate remains 0%, indicating complete failures in executing the full workflow. These results highlight a critical gap: current agents lack the context propagation and multi-stage planning capabilities required to chain individual DevOps tasks into cohesive workflows that mirror real-world deployment pipelines.

## 5 CONCLUSION AND FUTURE WORKS

We present DEVOPS-GYM, a comprehensive benchmark that evaluated agentic systems across the complete DevOps cycle through four critical stages. The evaluation results revealed substantial limitations in current agentic systems, with agents demonstrating particularly poor performance on monitoring and build and configuration tasks, highlighting a critical disconnect between current AI capabilities and real-world DevOps requirements. For the future, first, we will follow our proposed methodology to construct more tasks in each stage, covering more error types as well as broader projects. Second, we will also extend DEVOPS-GYM to other programming languages and more DevOps stages. Finally, we call for community-wide efforts to advance full DevOps automation by enriching DEVOPS-GYM with additional scenarios and metrics, developing more capable agents, and training specialized models optimized for agent workflows.

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

## A    CLARIFICATION ON THE USE OF LLMS

The authors acknowledge the use of large language models, employed exclusively to assist with grammar correction, proofreading, and minor stylistic refinements throughout the manuscript. The use of LLMs was strictly limited to language polishing and did not contribute to the research content itself. All core research, methodology, experimental design, and substantive findings represent the original work of the authors, who retain full responsibility for the content and conclusions presented.

## B    DATASET CONTAMINATION PREVENTION

To prevent data contamination from pre-training corpora, we implement a systematic detection protocol to select repositories for DEVOPS-GYM. For each candidate repository, we sample 20 unique code snippets from CI/CD configurations, build files, and test files, which are domain-specific artifacts less likely to appear in general pre-training data. Following the prefix completion approach (Carlini et al., 2021; Staab et al., 2023), we first randomly select a start point, then extract a 50-token prefix from each snippet and prompt the model to generate a continuation. We then assess contamination by comparing the model's 50-token prediction $a$ with the ground truth suffix $g$ from the original code.

We measure similarity using five complementary metrics: (1) normalized Levenshtein distance ratio $s$ (Lcvenshtcin, 1966), (2) consecutive matching token count from the prefix $t$, (3) position of the first token mismatch $p$, (4) longest common substring length $l$ measured in tokens (Gusfield, 1997), and (5) sentence-level BLEU score $b$ (Papineni et al., 2002). A snippet is classified as high-risk if any metric exceeds its respective threshold: $s > 0.7$, $t > 30$, $p > 30$, $l > 30$, or $b > 0.5$. These thresholds are calibrated to identify cases where the model demonstrates suspiciously accurate reproduction, indicating potential memorization. The repository-level contamination rate $C$ is defined as the proportion of high-risk snippets among the 20 samples.

The results for different repositories are shown in Table 2. To ensure evaluation integrity while maintaining sufficient dataset scale, we exclude repositories with $C \geq 0.2$ and retain only those with LOW or LOW-MODERATE contamination levels. This protocol ensures that our benchmark measures genuine reasoning and problem-solving capabilities rather than memorized patterns from pre-training. The 20% threshold balances contamination control with dataset scale. Our analysis shows that even the cleanest repositories (e.g., junit-framework) exhibit 10% contamination due to unavoidable surface-level similarities in public code (e.g., common API patterns, standard configurations). The 20% threshold ensures that 80%+ of code snippets show no memorization while maintaining sufficient scale for meaningful evaluation.

To prevent data leakage through repository history, we implement strict isolation measures. Following reports of contamination in SWE-bench where agents could access future repository states containing solutions or implementation hints (Kahn, 2025), we sanitize all repositories by removing git metadata and provide only the codebase state at the point of issue creation, ensuring agents cannot query future commits and branches containing fixes.

## C    TASK DETAILS FOR ISSUE RESOLVING AND TEST GENERATION

### C.1    ISSUE RESOLVING

In typical DevOps workflows, once problems are identified, whether through monitoring alerts, user reports, test failures, or system crashes, engineers must translate these diverse diagnostic inputs into code fixes. This task evaluates agents' ability to resolve bugs regardless of their discovery method. While issue resolving has been explored in benchmarks like SWE-bench, its inclusion here is essential for comprehensive DevOps evaluation, as fixing code remains a core responsibility whether the bug was found through sophisticated monitoring or a simple error log. Agents must understand existing codebases, locate bugs based on provided descriptions, and implement minimal fixes that resolve issues while preserving the normal functionalities.

- **Input:** (i) A buggy repository; (ii) Natural language bug description; (iii) Development tools (`git`, `grep`, `sed`, `awk`) and language-specific test runners.
- **Expected Output:** Patch file in unified diff format that resolves the issue.
- **Ground Truth:** Developer-provided fixes from merged GitHub pull requests or expert-written patches for injected bugs.
- **Evaluation:** Pass the Fail-to-pass test and didn't introduce new fails.
- **Data Collection:** We adapt the Multi-SWE-bench Zan et al. (2025) collection pipeline for data collection. The pipeline consists of three stages: (i) *repository selection:* we target well-maintained Java/Go projects with comprehensive test suites and active development

Table 2: Contamination risk analysis for different repositories across LLMs. The highest contamination rate for each repository is marked as **bold**.

| Repository | Max Contamination | Avg Contamination | GPT-4o | Claude-Sonnet-4 |
|---|---|---|---|---|
| act | 0.2000 | 0.1000 | 0.0000 | **0.2000** |
| beego | 0.5625 | 0.2812 | 0.0000 | **0.5625** |
| caddy | 0.2667 | 0.2000 | 0.1333 | **0.2667** |
| checkstyle | 0.5000 | 0.4583 | 0.4167 | **0.5000** |
| echo | 0.2000 | 0.1000 | 0.0000 | **0.2000** |
| etcd | 0.2222 | 0.1389 | 0.0556 | **0.2222** |
| fastjson2 | 0.7333 | 0.5667 | 0.4000 | **0.7333** |
| frp | 0.2500 | 0.1250 | 0.0000 | **0.2500** |
| fzf | 0.1875 | 0.1562 | **0.1875** | 0.1250 |
| gin | 0.4000 | 0.2667 | 0.1333 | **0.4000** |
| go-zero | 0.1765 | 0.1471 | 0.1176 | **0.1765** |
| gorm | 0.2500 | 0.1562 | 0.0625 | **0.2500** |
| hugo | 0.1250 | 0.0625 | 0.0000 | **0.1250** |
| istio | 0.3000 | 0.2000 | 0.1000 | **0.3000** |
| junit-framework | 0.1000 | 0.1000 | 0.1000 | 0.1000 |
| lazygit | 0.2308 | 0.1154 | 0.0000 | **0.2308** |
| logstash | 0.2222 | 0.1944 | 0.1667 | **0.2222** |
| mockito | 0.4000 | 0.3000 | 0.2000 | **0.4000** |
| spotbugs | 0.3333 | 0.2778 | 0.2222 | **0.3333** |

(commits within 6 months), (ii) *PR filtering:* we select pull requests that resolve GitHub issues and include test modifications, indicating test-validated fixes, and (iii) *execution validation:* we verify that associated tests demonstrate fail-to-pass transitions, where tests fail on the pre-patch codebase and succeed after patch is applied, thereby ensuring both issue reproducibility and patch correctness.

## C.2 TEST GENERATION

The third stage ensures that resolved issues cannot recur by creating comprehensive unit tests. While test generation has been explored in prior work Mündler et al. (2024), its inclusion here serves the DevOps pipeline where fixes must be validated before deployment. Our setting mirrors real-world scenarios where test developers cannot assume patch correctness. Agents must generate tests independently based solely on bug descriptions and source code examination, without access to the proposed fixes. This approach reflects common DevOps practice where unit tests serve as independent validation of patch quality rather than mere confirmation of known solutions. Furthermore, this constraint also requires agents to fully understand the bug's root cause from description alone and create tests that would have caught the original issue, particularly challenging for performance and resource issues identified through monitoring, which require specialized test assertions beyond functional correctness.

- **Input:** (i) Buggy repository; (ii) Bug description; (iii) File manipulation tools (`touch`, `echo`, `cat`) and test frameworks. Following established settings (Mündler et al., 2024), agents receive only the buggy repository and issue description without access to the ground truth patch or fix implementation.

- **Expected Output:** Patch file for the test in diff format (e.g., `pytest`, `JUnit`) that captures the bug's behavior.

- **Ground Truth:** Developer-provided tests from GitHub PRs (when available) or expert-written regression tests validated to detect the specific bug.

- **Evaluation:** Following SWT-Bench Mündler et al. (2024), we measure test effectiveness through fail-to-pass transitions. A test generation attempt succeeds when: (i) at least one

Table 3: Evaluation results on end-to-end tasks for different agent frameworks and different LLMs. The best result for each stage are marked as **bold**.

| Repository/Task | Model | Build & Config | Monitoring | Issue Resolving | Test Generation |
|---|---|---|---|---|---|
| minio_cpu-usage | Claude-4-Sonnet | 1 | 0 | 1 | 1 |
| | GPT-5-mini | 0 | 0 | 0 | 0 |
| minio_file-handle-leak | Claude-4-Sonnet | 0 | 0 | 0 | 0 |
| | GPT-5-mini | 0 | 0 | 0 | 0 |
| minio_io-latency | Claude-4-Sonnet | 0 | 0 | 0 | 0 |
| | GPT-5-mini | 0 | 0 | 0 | 0 |
| minio_memory-leak | Claude-4-Sonnet | 0 | 1 | 0 | 0 |
| | GPT-5-mini | 0 | 0 | 0 | 0 |
| pocketbase_cpu-usage | Claude-4-Sonnet | 1 | 0 | 0 | 0 |
| | GPT-5-mini | 1 | 0 | 0 | 0 |
| pocketbase_file-handle-leak | Claude-4-Sonnet | 0 | 0 | 0 | 0 |
| | GPT-5-mini | 1 | 0 | 0 | 0 |
| pocketbase_io-latency | Claude-4-Sonnet | 1 | 0 | 0 | 0 |
| | GPT-5-mini | 1 | 0 | 0 | 0 |
| pocketbase_memory-leak | Claude-4-Sonnet | 1 | 1 | 0 | 0 |
| | GPT-5-mini | 1 | 0 | 0 | 0 |
| spring-petclinic_cpu-usage | Claude-4-Sonnet | 1 | 0 | 1 | 0 |
| | GPT-5-mini | 1 | 0 | 0 | 0 |
| spring-petclinic_file-handle-leak | Claude-4-Sonnet | 1 | 0 | 1 | 0 |
| | GPT-5-mini | 1 | 0 | 0 | 0 |
| spring-petclinic_memory-leak | Claude-4-Sonnet | 1 | 0 | 0 | 0 |
| | GPT-5-mini | 1 | 1 | 1 | 0 |
| tidb_file-handle-leak | Claude-4-Sonnet | 1 | 1 | 0 | 1 |
| | GPT-5-mini | 1 | 1 | 0 | 0 |
| tidb_memory-leak | Claude-4-Sonnet | 0 | 1 | 0 | 0 |
| | GPT-5-mini | 1 | 0 | 0 | 1 |
| tidb_sql-handle-leak | Claude-4-Sonnet | 1 | 1 | 0 | 0 |
| | GPT-5-mini | 1 | 1 | 0 | 0 |

generated test fails on the original buggy codebase and passes after patch application (fail-to-pass transition), and (ii) no tests exhibit failures on the patched code. This metric ensures that generated tests accurately capture the bug's behavior while avoiding false positives. We report the success rate as the percentage of tasks where agents produce valid fail-to-pass tests without introducing spurious failures.

- **Data Collection:** Test generation tasks are derived from the same issues collected for issue resolving phase. For each issue resolving task, we create a corresponding test generation task.

## D  EXPERIMENTAL SUPPLEMENT

### D.1  GENERATION AND ENVIRONMENT.

We use vLLM (Kwon et al., 2023) to host Qwen3-Coder-30B on our local server, and all other models are accessed via their official APIs. For the hyper-parameters of model settings, we employ a temperature of 0.7 and top-p sampling of 0.95, with a maximum context length of 256K tokens. Additionally, we implement a 60-second timeout with up to 3 retry attempts to ensure robust inference. All experiments are executed within the isolated and reproducible Docker container environments provided by DEVOPS-GYM.

### D.2  ERROR ANALYSIS FOR MONITORING TASKS

We also identified four common error types for monitoring tasks from the agent run logs. First, inadequate monitoring methodology (37%): Agents used one-time commands (e.g., `top`) instead

Table 4: Performance across five independent runs on a sampled subset of 50 tasks.

| Agent | Model | Round1 | Round2 | Round3 | Round4 | Round5 | Mean(STD) | Pass@5 |
|---|---|---|---|---|---|---|---|---|
| Claude Code | Claude-4-Sonnet | 16.00% | 16.00% | 20.00% | 20.00% | 22.00% | 18.80% (2.40) | 26.00% |
| OpenHands | o4-mini | 14.00% | 16.00% | 18.00% | 20.00% | 16.00% | 16.80% (2.04) | 20.00% |

of continuous monitoring (e.g., `watch -n 1`), occasionally yielding coincidental correct results on error-free instances. Second, pemature conclusion (26%): Agents submitted answers without performing monitoring or completing diagnostic procedures. Third, insufficient temporal granularity (11%): Agents monitored correctly but used overly coarse sampling intervals (10-60s), missing transient anomalies like CPU spikes. Lastly, interpretation failure (26%): Agents collected metrics correctly but failed to analyze them accurately or provided no analysis.

### D.3 STABILITY OF AGENT PERFORMANCE ACROSS MULTIPLE INDEPENDENT RUNS

To evaluate the stability of agent performance, we randomly sampled 50 tasks from DEVOPS-GYM and executed two representative agent–model pairs—Claude Code + Claude-4-Sonnet and Open-Hands + o4-mini—across five independent runs. Table 4 summarizes the results. Both agents exhibit consistent single-run performance, with Claude Code ranging between 16–22% and OpenHands between 14–20%. The mean accuracies across the five runs are 18.8% and 16.8% with the standard deviation of 2.40 and 2.04, respectively, indicating that run-to-run variance is relatively small. When aggregating over the five trajectories, success rates increase modestly (Claude Code: 26%, Open-Hands: 20%), but this improvement does not materially affect the relative ranking of the agents or the conclusions of our study. These findings confirm that DEVOPS-GYM yields stable results and that model performance is not sensitive to randomness in agent execution.

## E CASE STUDY

### E.1 BUILD

This case shows that the agent fails to properly handle test execution failures while still achieving the core migration objectives.

---

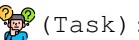(Task):

You are tasked with migrating the MobArena plugin build system from Maven to Gradle while maintaining functional equivalence.

The environment contains:

A Maven project at /app/mobarena/ – this is your starting point

Build artifacts and logs from the Maven build in /app/results/old_build/ and /app/results/old-.log

Your task is to:

Start with the Maven project at /app/mobarena/

Create a complete Gradle build configuration (build.gradle, gradle.properties, etc.)

Migrate all Maven functionality to Gradle equivalents

Ensure the migrated build maintains functional equivalence with the original Maven build

Success criteria:

---

```
The Gradle build completes successfully with ./gradlew clean
build

All tests pass with ./gradlew test

The generated JAR is functionally equivalent to the original
Maven JAR

Build produces the same core artifacts as the Maven version
```

🤖 (Agent Action):

I'll start by examining the Maven project structure and
understanding the current build configuration to plan the
migration to Gradle.

function=codebase_search

query="What is the current Maven build configuration and
project structure?"

target_directories=["/app/mobarena"]

🤖 (Agent Action):

Now I need to understand the Maven build configuration by
examining the pom.xml file to see what dependencies, plugins,
and build settings need to be migrated.

function=read_file

target_file="/app/mobarena/pom.xml"

[... extensive migration work creating build.gradle,
configuring dependencies, setting up Gradle wrapper ...]

🤖 (Agent Action):

Now let me run the tests to see if they all pass as expected.

function=bash

command="cd /app/mobarena && ./gradlew test --info | grep -E
(Test|PASSED|FAILED|SKIPPED|BUILD)"

The tests are running but many are failing.  This seems to be
related to missing runtime dependencies and mocking issues.
BUILD FAILED in 6s 253 tests completed, 88 failed

⇒ {Comments:  Agent successfully migrates the build system
and produces correct artifacts, but fails to resolve all test
failures.  The core issue is that while the agent achieves
the primary migration goals (successful build, correct JARs,
functional equivalence), it doesn't ensure all tests pass,
which was explicitly listed as a success criterion.}

This case shows that the agent fails to include Caddy-specific build tags in the GoReleaser configuration.

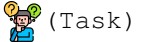 (Task):

```
You are tasked with setting up GoReleaser for the Caddy project
to enable multi-platform releases.
```

The environment contains:

A Go project at /app/caddy-project - this is your starting point

Build artifacts and logs from the basic Go build in /app/results/old_build/ and /app/results/old-.log

GoReleaser is pre-installed and available in the PATH

Your task is to:

Start with the basic Go project at /app/caddy-project

Create a .goreleaser.yml configuration file with modern GoReleaser syntax

Configure multi-platform builds for Linux, Windows, macOS, and FreeBSD

Set up cross-compilation for multiple architectures (amd64, arm, arm64, s390x, ppc64le, riscv64) with appropriate platform exclusions

Configure archive generation with proper platform naming

Configure build tags and ldflags for optimal binary generation

The configuration should include:

Cross-platform builds with CGO disabled for static binaries

Build configuration for cmd/caddy/main.go structure

Archive replacements for clean platform naming (macOS, Linux, Windows)

Windows-specific ZIP format override

Changelog generation with proper filtering

Checksum configuration with SHA512 algorithm for release integrity

Release configuration with draft mode enabled to prevent accidental releases

Success criteria:

GoReleaser configuration validates successfully with goreleaser check

Snapshot build completes with goreleaser build --snapshot --clean

Multiple platform binaries are generated (Linux, Windows, macOS, FreeBSD)

Generated binaries maintain original functionality

Archive naming follows GoReleaser conventions

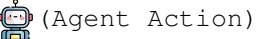(Agent Action):

I need to start by analyzing the codebase structure to understand the Go project layout and then examine any existing build artifacts to understand the current build process.

```
cd /app/caddy-project && find .  -type f -name "*.go" | head
-20
```
(Agent Action):

Good, I can see this is a Caddy project with the main entry
point at ./cmd/caddy/main.go.  Now let me examine the project
structure more thoroughly and check if there are any existing
configuration files.

```
cd /app/caddy-project && ls -la && echo "=== Checking for
existing config files ===" && find .  -name "*.yml" -o -name
"*.yaml" -o -name "go.mod" -o -name "Makefile" | head -10
```

...

(Agent Action):

Now I understand the project structure and version handling.
Caddy uses Go modules for version information, so I don't
need custom ldflags for version injection.  Let me create a
comprehensive GoReleaser configuration file that meets all the
requirements.

```yaml
cd /app/caddy-project && cat <<'EOF' > .goreleaser.yml

# GoReleaser configuration for Caddy

# This configuration enables multi-platform releases with
proper cross-compilation

project_name:  caddy

before:

hooks:

# Clean up any previous builds

- go mod tidy

- go mod download

builds:

- id:  caddy

# Build from the main entry point

main:  ./cmd/caddy/main.go

binary:  caddy

# Disable CGO for static binaries

env:

- CGO_ENABLED=0

# Build flags for optimization

flags:

- -trimpath

# Linker flags for smaller binaries

ldflags:

- -s -w
```

```
- -extldflags "-static"
# Multi-platform builds
goos:
- linux
- windows
- darwin
- freebsd
# Multi-architecture builds with appropriate exclusions
goarch:
- amd64
- arm
- arm64
- s390x
- ppc64le
- riscv64
...
EOF
⇒ {Comments:  The agent creates a comprehensive GoReleaser
configuration but fails to include Caddy-specific build tags
(nobadger, nomysql, nopgx) that are essential for proper Caddy
builds.  The configuration validates syntactically but is
functionally incomplete for the Caddy project.}
```

## E.2  MONITORING

This case shows that the agent fails to monitor the system continuously.

```
👥(Task):
You are tasked with monitoring a server program execution to
identify any potential unusual patterns and potential issues.

Your environment contains:

- A server setup:  a server is running.

- Standard system monitoring tools (curl, ls, top, ps, htop,
watch, etc.)

Your task is to:

1) Monitor the system behavior (e.g., server network requests,
system resource usage) over time,

2) Use standard system monitoring tools to observe system
behavior,

3) Identify any unusual patterns

...
```

> 🤖(Agent Action):
>
> I'll monitor the minio server system behavior over 8 minutes to identify resource usage patterns and potential issues. Let me start by setting up a comprehensive monitoring approach.
>
> function=TodoWrite content="1. Check initial system state, 2. Setting up continuous monitoring for CPU, memory, and disk usage, 3. Monitor network connections and file descriptors, 4. Monitor I/O operations and latency."
>
> 🤖(Agent Action):
>
> Now let me observe the system.
>
> ls -la /proc/14/fd
>
> cat /proc/14/io
>
> ⇒ {Comments: Agent should monitor the system continuously, e.g., watch -n 1 cat /proc/14/io}
>
> ⚙️(Tool Execution Outputs):
>
> ...

This case demonstrates that agent wrongly get a final conclusion due to limited context length.

> 🧑‍🤝‍🧑(Task):
>
> You are tasked with monitoring a server program execution to identify any potential unusual patterns and potential issues.
>
> Your environment contains:
>
> – A server setup: a server is running.
>
> – Standard system monitoring tools (curl, ls, top, ps, htop, watch, etc.)
>
> Your task is to:
>
> 1) Monitor the system behavior (e.g., server network requests, system resource usage) over time,
>
> 2) Use standard system monitoring tools to observe system behavior,
>
> 3) Identify any unusual patterns
>
> ...
>
> 🤖(Agent Action):
>
> I'll monitor the server system behavior.
>
> watch -n 1 "echo '=== $(date) ==='; top -b -n 1 | head -20" >> /monitor_cpu.log
>
> watch -n 1 "echo '=== $(date) ==='; free -h; echo ''; ps aux --sort=-%mem | head -10" >> /monitor_memory.log
>
> watch -n 3 "echo '=== $(date) ==='; echo 'Open FDs:'; ls /proc/1/fd 2>/dev/null | wc -l; lsof -p 1 2>/dev/null | wc -l" >> /monitor_fd.log

Table 5: DevOpsBench Dataset Statistics

| Stage | Languages | # Tasks | Repo Num | Avg. Repo Files | Avg. Human Time(min) |
|---|---|---|---|---|---|
| Build & Config | Java&Go | 48 | 26 | 1579 | 53 |
| Monitoring | Java&Go | 30 | 7 | 3343 | 21 |
| Issue Resolving | Java&Go | 310 | 16 | 2162 | 35 |
| Test Generation | Java&Go | 310 | 16 | 2162 | 45 |

Table 6: Evaluation Results on different error types of build implementation

| Agent | Model | Migration | Target Release | Plugin Integration | Version Upgrades |
|---|---|---|---|---|---|
| OpenHands | Qwen3-Coder-30B | 1/8 | 0/5 | 0/2 | 1/3 |
| | o4-mini | 1/8 | 0/5 | 1/2 | 1/3 |
| | DeepSeek-V3.1 | 0/8 | 0/5 | 0/2 | 1/3 |
| | Gemini-2.5-Pro | 0/8 | 0/5 | 0/2 | 1/3 |
| | Claude-4-Sonnet | 3/8 | 0/5 | 0/2 | 1/3 |
| mini-SWE-Agent | Claude-4-Sonnet | 2/8 | 1/5 | 1/2 | 2/3 |
| Claude Code | Claude-4-Sonnet | 2/8 | 1/5 | 1/2 | 2/3 |

```
...
(Agent Action):  Now let me check the CPU log file to detect
potential system anormalies.
cat /monitor_cpu.log
(Tool Execution Outputs):

(CPU log files)...
(Agent Action):

This session is being continued from a previous conversation
that ran out of context.  The conversation is summarized below:

...
(Agent Action):

According to the summary I can see that CPU activity is
abnormal.  So the answer is cpu usage.
⇒ {Comments:  Agent should check other log files.  It
successfully monitor the system but fail to analyze the
results.}
```

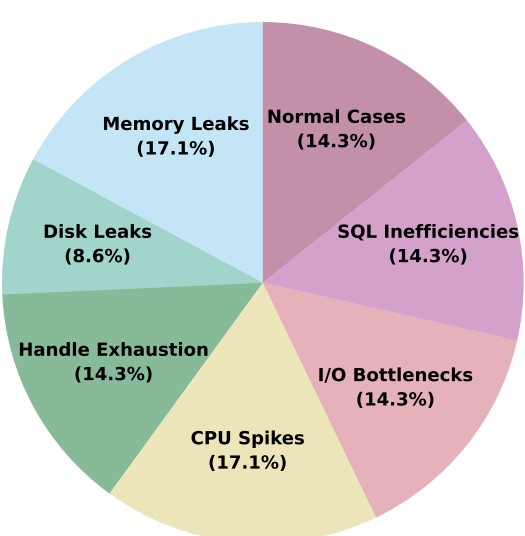

Figure 3: Monitoring anomaly distribution.

