# OpenReview forum: "DevOps-Gym: Benchmarking AI Agents in Software DevOps Cycle"
_ICLR.cc/2026/Conference — ICLR 2026 Poster_

### Official Review · Reviewer_jCaQ · 2025-10-29

**Soundness:** 3
**Presentation:** 2
**Contribution:** 3
**Rating:** 8
**Confidence:** 4

**Summary:**

The paper proposes a benchmark for evaluating SWE Agents on four different activities frequently encountered in the software development cycle: build/configuration, monitoring, bug fixing and test generation. They source and vet tasks for Java and Go and evaluate across popular coding agents and models. Their analysis shows that current systems fall short on configuration and monitoring tasks.

**Strengths:**

The paper presents a novel perspective to evaluating coding agents. The tasks presented in the paper go beyond popular bug-fixing setting and are sourced from Java and Go projects as opposed to the popular python setting. Especially the monitoring tasks which requires the system to interact with the environment over time. The difference in performance between agents also indicates the tasks to be a good discriminator. Evaluation with this benchmark maybe useful to practitioners working on industrial codebases that require maintenance beyond fixing bugs.

**Weaknesses:**

1. Two of the four tasks are not novel: bug-fixing and test generation are well explored in existing literature.

2. While the paper presents useful contributions to practitioners who may want to evaluate SWE-agents for DevOps use cases, the paper offers limited insight into the behaviour of current agent frameworks or models, limiting usefulness to agent and model developers and hence relevance to this venue.

**Questions:**

1. What are guards against agent finding trivial ways to resolve tasks. For example in the build/configuration task, what if the agent modifies the build scripts to pass trivially?

2. How is evaluation performed for the monitoring task. The paper mentions that the LLM has to identify the type of anomaly? How is the response from the LLM evaluated to determine correctness?

---

> ### Author Response · Authors · 2025-11-22
> **Rebuttal by Authors**
>
> We thank the reviewer for the positive, insightful, and constructive feedback. Please refer to the following contents for our responses.
>
> ## Q1. More insight into the behaviour of current agent frameworks and models
>
> We conducted detailed error analysis for build and monitor tasks and summarized common error types made by current agents. Details can be found in revised manuscript (Sec 4.2).
>
> For Build/Config tasks, we diagnose three main error patterns: 1) Toolchain and environment instrumentation limitations (33%). Agents cannot validate configuration artifacts (e.g., unused imports, missing dependencies) because necessary validators are unavailable in the environment. 2) Multi-step reasoning failures (23\%). Agents resolve initial errors but lose track of remaining issues due to context preservation limitations and lack of iterative "fix-run-verify" loops. 3) Domain-specific knowledge gaps (37\%). These failures reflect technical complexity beyond current model capabilities, including build-system migrations, type compatibility issues, and platform-specific constraints.
>
> For monitoring, error analysis reveals four failure modes: 1) Inadequate monitoring methodology (37%): Agents use one-time commands instead of continuous observation, failing to track behavior over time. 2) Premature conclusion without monitoring (26%): Agents submit answers without executing any diagnostic commands. 3) Insufficient temporal granularity (11%): Agents use overly coarse sampling intervals (10-60s), missing transient anomalies like CPU spikes. 4) Interpretation failure (26%): Agents collect metrics correctly but fail to analyze them or draw incorrect conclusions.
> There are no false positives for monitor tasks. However, we observe that a substantial portion of seemingly correct predictions stem from the first error type: agents using inadequate one-time observations (e.g., a single top command) coincidentally yield correct answers on baseline instances with no injected errors. Since these instances represent normal system states, even a single snapshot can appear normal, leading agents to correctly conclude "no issues detected" despite employing improper monitoring methodology that would fail to detect actual anomalies.
>
> ## Q2. How to prevent shortcut solutions?
>
> We prevent shortcuts in two ways.
> (1) Task prompts explicitly require agents to fix the issue or complete the requirements correctly, not modify scripts to bypass failures.
> (2) Evaluation checks functional equivalence, not just a successful build. For example, in migration tasks for build/config, we compare generated artifacts (e.g., binaries, dependency outputs) against the ground truth, so any trivial script modification will fail validation.
>
> ## Q3. How is evaluation performed for the monitoring task?
>
> We have clarified the evaluation for the monitoring task in the revised manuscript.
> In the prompt, we define the five anomaly types and explicitly instruct the model to write its diagnosis into a specified file in a single line without any explanation. Evaluation is performed using automated pytest scripts that check: (1) whether the diagnosis file exists, and (2) whether the diagnosed anomaly type matches the ground truth.

---

> ### Author Response · Authors · 2025-11-26
> **Following up on our response**
>
> Dear Reviewer jCaQ,
>
> Thank you for your valuable feedback on our manuscript. As the discussion period draws to a close, we wanted to follow up on our previous responses to your comments. We would greatly appreciate knowing whether our revisions and clarifications have adequately addressed your concerns, or if there are any remaining questions we can help resolve. We remain available to provide any additional information or clarification that would be helpful.
>
> Best,
>
> Authors

---

### Official Review · Reviewer_NdRZ · 2025-10-31

**Soundness:** 3
**Presentation:** 4
**Contribution:** 3
**Rating:** 6
**Confidence:** 5

**Summary:**

This paper presents a benchmark, DevOps-gym, for evaluating agents on end-to-end DevOps workflows. The authors curate a benchmark of nearly 700 tasks from 30+ repositories in Java and Go languages. The main contribution of the paper is in covering the entire spectrum of DevOps operations - build, monitor, fix and test. The benchmark comes with pre-set runtime environments and tools. The experimental results on a few agents and multiple LLMs show that they succeed to varying degrees across the task categories, and generally well-below the success rate on popular benchmarks like swe-bench.

**Strengths:**

- The paper makes a comprehensive contribution, covering all phases of DevOps operations. The benchmark is based on real-world repositories and targets languages common in industry like Java and Go (compared to benchmarks that target Python). The experimental setup is well-designed.
- The paper is written very nicely. The key aspects of design of the benchmark are explained properly, showing both the diversity of sub-tasks within each task categories, and clearly specifying input, output and evaluation metrics. The analysis of results is also insightful.
- The experimental evaluation is supportive of the value of this benchmark, by demonstrating that the current agents and LLMs are not very successful on this benchmark and their performance varies across task categories.

**Weaknesses:**

- The benchmark curation involved synthetic data generation where faults are injected by experts. The paper claims that they are inspired by real-world scenarios, but the adequate details are not provided.
- The paper uses OpenHands and mini-SWE-Agent as harnesses. Though they may allow access to the terminal, their tools are primarily designed for assisting agents in GitHub issue resolution. No effort is made to design an agentic harness that specifically aids the agent in DevOps tasks.
- It is well-known that test-time scaling can improve the performance of agents substantially. The paper does not conduct scaling experiments.
- The benchmark does not cover the CI/CD aspects of DevOps tasks and does not throw light on how these agents could operate in CI/CD pipelines with platform-specific tools, e.g., GitHub actions.
- Typo: interpreting their returns

**Questions:**

- How many tasks out of 698 are real and how many are synthetic? What is the distribution by problem types (e.g., memory leaks, build failure, etc.) between real vs synthetic tasks? What is the reason those cannot be found naturally in the wild if your task distribution is to capture the real-world distribution?
- How would the performance of the LLMs change if DevOps-specific agent harness is prepared and made available to them?
- How does the performance of the agents change with test-time scaling, e.g., sampling multiple trajectories?

---

> ### Author Response · Authors · 2025-11-22
> **Rebuttal by Authors**
>
> We thank the reviewer for the positive, insightful, and constructive feedback. Please refer to the following contents for our responses.
>
> ## Q1. Task distributions and why do we need sythesize tasks?
>
> During the rebuttal, we added four new monitoring tasks and six build tasks. These tasks also bring two new repos in Go and Java to our benchmark. We will continue expanding these categories over time. Out of 708 tasks, 659 are collected from real-world repositories and 49 are synthetic. The distribution by task type is: Issue Resolution (310 real, 0 synthetic), Test Generation (310 real, 0 synthetic), Build Automation (34 real, 20 synthetic), and Monitoring (5 real, 29 synthetic).
>
> The reason for having these synthetic data is as follows:
>
> Monitoring tasks evaluate an agent's ability to diagnose system-wide anomalies based on observable signals (logs, metrics, traces), regardless of whether the underlying issue originated naturally or was artificially induced. It makes synthetic and real-world tasks functionally equivalent. In fact, synthetic monitoring scenarios offer significant advantages: they provide better controllability to create clear, reproducible anomaly patterns (e.g., deliberate memory leaks, CPU spikes) and ensure consistent evaluation conditions.
>
> For build tasks, the 14 synthetic examples primarily cover build implementation scenarios (e.g., build from scratch requiring generation of build configuration files), which happen in real-world requirements but rarely appear as GitHub issues. GitHub issues predominantly document build failures rather than initial build setup tasks. The synthetic tasks fill this gap to ensure comprehensive coverage of the build automation spectrum.
>
> ## Q2. How would the performance of the LLMs change if DevOps-specific agent harness is prepared and made available to them?
> We believe performance will increase with a DevOps-specific harness, but our benchmark intentionally standardizes the agent interface to ensure fair comparison. Custom harness design is an important but separate research direction for future work.
>
> ## Q3. Results of sampling multiple trajectories
>
> To evaluate stability, we randomly choose 50 subset tasks from the benchmark and test with ClaudeCode + Claude-sonnet-4 and OpenHands + o4-mini for five independent runs.
> The table summarizes the results. Both agents exhibit consistent single-run performance, with Claude Code ranging between 16–22% and OpenHands between 14–20%. The mean accuracies across the five runs are 18.8% and 16.8% with the standard deviation of 2.40 and 2.04, respectively, indicating that run-to-run variance is relatively small. When aggregating over the five trajectories, success rates increase modestly (Claude Code: 26%, OpenHands: 20%), but this improvement does not materially affect the relative ranking of the agents or the conclusions of our study. These findings confirm that our benchmark yields stable results and that model performance is not sensitive to randomness in agent execution.
>
> |   | Round1 | Round2 | Round3 | Round4 | Round5 |   Mean (STD)  | Pass@5 |
> |---|---| --- | --- | --- | --- | --- | --- |
> | Claude Code + Claude-4-Sonnet | 16.00% | 16.00% | 20.00% | 20.00% | 22.00% | 18.80% (2.40) | 26.00% |
> | OpenHands + o4-mini      | 14.00% | 16.00% | 18.00% | 20.00% | 16.00% | 16.80% (2.04) | 20.00% |
>
>
>
> ## Q4. Testing-phase scaling
>
> The agents in our evaluation lack internal ranking mechanisms that support test-phase scaling for our tasks. For example, OpenHands' judgment model is specifically designed for SWE-bench. While we could employ a separate model as a judge, this would deviate from the original agent design.
>
> Instead, we sampled multiple trajectories and computed pass@k to assess the impact of test-time scaling. As shown in the table above, performance improves only marginally with pass@5, demonstrating the difficulty of DevOps-Gym tasks.
>
>
> ## Q5. Cover the CI/CD aspects of DevOps tasks
>
> As the first work to assess LLM's capabilities in the DevOps domain, our goal in DevOps-Gym is to cover some core workflows. Stages such as CI/CD automation, or infrastructure management strongly depend on mutable external systems (cloud APIs, Kubernetes clusters, pipeline triggers, remote state), which make both task creation and evaluation difficult. These steps primarily orchestrate build/monitor/resolve/test stages rather than introducing fundamentally new reasoning capabilities.
> In contrast, the four stages we selected—build & configuration, monitoring, issue resolving, and test generation—represent the core reasoning and tool-using capabilities required in DevOps automation while remaining feasible for rigorous, reproducible benchmarking.
> We agree the omitted stages are promising extensions, and we plan to expand some of the important DevOps stages like deployment and rollback, which can be connected with the current design. We have added clarifications on these design choices and limitations in the revised manuscript.

---

> > ### Comment · Reviewer_NdRZ · 2025-11-27
> > **Reviewer response**
> >
> > Thank you for clarifications, additional results and updates to the paper. I'm generally positive about the paper. I feel the main weakness is that the agent harnesses are not extended for the proposed tasks. The authors should acknowledge this limitation in the paper.

---

> ### Author Response · Authors · 2025-11-26
> **Following up on our response**
>
> Dear Reviewer NdRZ,
>
> Thank you for your valuable feedback on our manuscript. As the discussion period draws to a close, we wanted to follow up on our previous responses to your comments. We would greatly appreciate knowing whether our revisions and clarifications have adequately addressed your concerns, or if there are any remaining questions we can help resolve. We remain available to provide any additional information or clarification that would be helpful.
>
> Best,
>
> Authors

---

> ### Author Response · Authors · 2025-11-29
> **Author response**
>
> Thank you for your valuable feedback and positive assessment of our work. We acknowledge that extending agent harnesses  is an important direction. We will add this as future work in the revised paper, including plans to evaluate other agents (such as Treminus [1]) that operate exclusively through terminal interfaces. We will also explicitly acknowledge this limitation in the appropriate section.
>
> [1] https://www.tbench.ai/terminus

---

### Official Review · Reviewer_EL6m · 2025-10-31

**Soundness:** 3
**Presentation:** 3
**Contribution:** 3
**Rating:** 6
**Confidence:** 4

**Summary:**

This paper introduces DevOps-Gym, a benchmark for evaluating AI agents across the full DevOps cycle, covering build & configuration, monitoring, issue resolving, and test generation. The benchmark includes 698 tasks from 30+ real-world Java and Go repositories. Evaluation of state-of-the-art agents reveals poor performance across all stages, with particularly severe failures on monitoring (0-23.33%) and build tasks (12.5-58.33%). While this addresses a genuine gap in existing benchmarks, the execution has significant methodological weaknesses that limit the contribution's impact. There are some concerns outlined below.

**Strengths:**

1. **Addresses Real Gap:** First benchmark attempting end-to-end DevOps evaluation
2. **Realistic Task Design:** Docker environments with real tools better simulate practice than synthetic environments, and covering diverse DevOps stages (though incomplete)
3. **Extensive Manual Effort:** Authors invested significant time in task construction and validation
4. **Negative Results Are Valuable:** Showing current agents fail badly on DevOps tasks is an important finding
5. **Practical Relevance:** High potential impact for industrial applications

**Weaknesses:**

1. **No statistical rigor**: Missing confidence intervals, significance tests, multiple runs; single-run results unreliable given tiny task counts.
2. **Small dataset**: Only 30 monitoring and 48 build tasks, which is far smaller than benchmarks like SWE-bench, so this limits generalizability.
3. **Limited analysis**: The paper reports binary accuracy for monitoring, but there is no partial credit or false-positive analysis provided.
4. **Limited baselines**: Excludes major frameworks (Devin, Aider, etc.), so it's unclear if the results are significantly better than SOTA.

**Questions:**

1. **Statistical Rigor:** Why no confidence intervals, multiple runs, or significance tests? This is standard practice for benchmark papers. With small sample sizes (30 monitoring, 48 build tasks), single-run results are highly unreliable and prevent drawing meaningful conclusions about agent capabilities.

2. **Synthetic Task Validation:** Do you have some evidence that expert-injected bugs match real-world distributions? The work relies heavily on synthetic tasks for monitoring and build stages but doesn't provide validation that these reflect actual production issues. Without this validation, it seems incorrect to claim the benchmark represents "real-world" DevOps challenges?

3. **Ground Truth Agreement:** It might be good to include inter-annotator agreement scores. The paper mentions "three senior DevOps engineers independently validate" monitoring tasks and use multiple experts for other stages, but report on agreement metrics (Cohen's kappa, etc.). If experts disagree, then the validity of the canonical answer is questionable, which limits the benchmark credibility.

4. **Contamination Threshold:** Why allow up to 20% contamination instead of requiring something much smaller? The paper talks about decontamination procedures but then also includes repositories with substantial contamination. The standpoint is a bit confusing.

---

> ### Author Response · Authors · 2025-11-22
> **Rebuttal by Authors (Part 1)**
>
> We thank the reviewer for the positive, insightful, and constructive feedback. Please refer to the following contents for our responses.
>
> ## Q1. Adding statistical analysis through multiple runs
>
> To evaluate stability, we randomly choose 50 subset tasks from the benchmark and test with ClaudeCode + Claude-sonnet-4 and OpenHands + o4-mini for five independent runs.
> The table summarizes the results. Both agents exhibit consistent single-run performance, with Claude Code ranging between 16–22% and OpenHands between 14–20%. The mean accuracies across the five runs are 18.8% and 16.8% with the standard deviation of 2.40 and 2.04, respectively, indicating that run-to-run variance is relatively small. When aggregating over the five trajectories, success rates increase modestly (Claude Code: 26%, OpenHands: 20%), but this improvement does not materially affect the relative ranking of the agents or the conclusions of our study. These findings confirm that our benchmark yields stable results and that model performance is not sensitive to randomness in agent execution.
>
> |                               | Round1 | Round2 | Round3 | Round4 | Round5 |   Mean (STD)  | Pass@5 |
> | ----------------------------- | ------ | ------ | ------ | ------ | ------ | ------------- | ------ |
> | Claude Code + Claude-4-Sonnet | 16.00% | 16.00% | 20.00% | 20.00% | 22.00% | 18.80% (2.40) | 26.00% |
> |      OpenHands + o4-mini      | 14.00% | 16.00% | 18.00% | 20.00% | 16.00% | 16.80% (2.04) | 20.00% |
>
>
>
> ## Q2. Only 30 monitoring and 48 build implement tasks
>
> Monitoring and build/config tasks are inherently low-frequency but high-complexity in real-world systems. Unlike issue-resolving and test-generation tasks that can be automatically mined at scale, reproducing operational anomalies or build-chain tasks requires expert efforts and validation. Each instance carries significantly higher information density and diagnostic complexity, so the number of these two kind of tasks is relatively small.
> Notably, DevOps-Gym is designed as an evolving benchmark. We have recently added 10 new monitoring/build tasks during this rebuttal time and will continue expanding these categories over time.
> We hjave clarified the planned extensions in the revised manuscript (Sec 5).
>
> ## Q3. Provide detailed analysis for monitor and build
>
> We conducted detailed error analysis for build and monitor tasks and summarized common error types made by current agents. Details can be found in revised manuscript (Sec 4.2).
>
> For Build/Config tasks, we diagnose three main error patterns: 1) Toolchain and environment instrumentation limitations (33%). Agents cannot validate configuration artifacts (e.g., unused imports, missing dependencies) because necessary validators are unavailable in the environment. 2) Multi-step reasoning failures (23\%). Agents resolve initial errors but lose track of remaining issues due to context preservation limitations and lack of iterative "fix-run-verify" loops. 3) Domain-specific knowledge gaps (37\%). These failures reflect technical complexity beyond current model capabilities, including build-system migrations, type compatibility issues, and platform-specific constraints.
>
> For monitoring, error analysis reveals four failure modes: 1) Inadequate monitoring methodology (37%): Agents use one-time commands instead of continuous observation, failing to track behavior over time. 2) Premature conclusion without monitoring (26%): Agents submit answers without executing any diagnostic commands. 3) Insufficient temporal granularity (11%): Agents use overly coarse sampling intervals (10-60s), missing transient anomalies like CPU spikes. 4) Interpretation failure (26%): Agents collect metrics correctly but fail to analyze them or draw incorrect conclusions.
> There are no false positives for monitor tasks. However, we observe that a substantial portion of seemingly correct predictions stem from the first error type: agents using inadequate one-time observations (e.g., a single top command) coincidentally yield correct answers on baseline instances with no injected errors. Since these instances represent normal system states, even a single snapshot can appear normal, leading agents to correctly conclude "no issues detected" despite employing improper monitoring methodology that would fail to detect actual anomalies.
>
>
> ## Q4. Add new baselines
>
> We have now added an evaluation of agent Aider, one of the requested baselines, and updated the paper accordingly. Across all stages, Aider shows lower performance compared with stronger agent frameworks such as OpenHands and Claude Code (e.g., 6.25% on Build & Config vs. 47.91–58.33%, 9.67% on Issue Resolving vs. 23.87%). The  details are already updated in the paper revision.

---

> ### Author Response · Authors · 2025-11-22
> **Rebuttal by Authors (Part 2)**
>
> ## Q5. Synthetic task v.s. real-world tasks
>
> During the rebuttal, we added four new monitoring tasks and six build tasks. These tasks also bring two new repos in Go and Java to our benchmark. We will continue expanding these categories over time. Out of 708 tasks, 659 are collected from real-world repositories and 49 are synthetic. The distribution by task type is: Issue Resolution (310 real, 0 synthetic), Test Generation (310 real, 0 synthetic), Build Automation (34 real, 20 synthetic), and Monitoring (5 real, 29 synthetic).
>
> The reason for having these synthetic data is as follows:
>
> Monitoring tasks evaluate an agent's ability to diagnose system-wide anomalies based on observable signals (logs, metrics, traces), regardless of whether the underlying issue originated naturally or was artificially induced. It makes synthetic and real-world tasks functionally equivalent. In fact, synthetic monitoring scenarios offer significant advantages: they provide better controllability to create clear, reproducible anomaly patterns (e.g., deliberate memory leaks, CPU spikes) and ensure consistent evaluation conditions.
>
> For build tasks, the 14 synthetic examples primarily cover build implementation scenarios (e.g., build from scratch requiring generation of build configuration files), which happen in real-world requirements but rarely appear as GitHub issues. GitHub issues predominantly document build failures rather than initial build setup tasks. The synthetic tasks fill this gap to ensure comprehensive coverage of the build automation spectrum.
>
>
> ## Q6. Ground truth clarification
>
> Ground truth is not derived from subjective annotation. All tasks—including monitoring and build/configuration—have execution-based, deterministic ground truth. The role of multiple experts was solely to ensure reproducibility of the environment and verify that each anomaly is observable using the provided tools—not to “label” tasks.
>
> ## Q7.  Why allow 20% contamination in contamination detection?
>
> Thank you for this important question. We have updated Appendix B with additional clarification.
>
> The 20% threshold balances contamination control with dataset scale. Our analysis shows that even the cleanest repositories (e.g., junit-framework) exhibit ~10% contamination due to unavoidable surface-level similarities in public code (e.g., common API patterns, standard configurations). Contamination below 20% primarily reflects such partial matches rather than verbatim memorization. Setting a stricter threshold (e.g., C < 0.1) would reduce the dataset to only 3 repositories, severely limiting benchmark diversity. The 20% threshold ensures that 80%+ of code snippets show no memorization while maintaining sufficient scale for meaningful evaluation.

---

> ### Author Response · Authors · 2025-11-26
> **Following up on our response**
>
> Dear Reviewer EL6m,
>
> Thank you for your valuable feedback on our manuscript. As the discussion period draws to a close, we wanted to follow up on our previous responses to your comments. We would greatly appreciate knowing whether our revisions and clarifications have adequately addressed your concerns, or if there are any remaining questions we can help resolve. We remain available to provide any additional information or clarification that would be helpful.
>
> Best,
>
> Authors

---

### Official Review · Reviewer_uo9F · 2025-11-03

**Soundness:** 2
**Presentation:** 2
**Contribution:** 2
**Rating:** 4
**Confidence:** 5

**Summary:**

This paper presents DEVOPS-GYM, a benchmark created to test how well AI agents can handle the full software DevOps cycle, including building and configuration, monitoring, bug fixing, and test generation. It goes beyond previous benchmarks that focus only on coding or bug fixing by including real DevOps tasks that require tool use, environment setup, and step-by-step decision making. The dataset contains 698 tasks collected from over 30 open-source projects in Java and Go, combining real GitHub issues with tasks designed by experts to cover common DevOps problems.

The benchmark was built with attention to realism and fairness. The authors manually recreated build environments, monitored runtime behavior, and avoided data contamination from existing model training data. Each task type evaluates a different part of the DevOps process: building tasks test if the agent can fix dependency or build errors, monitoring tasks test detection of issues like memory leaks or CPU spikes, issue-resolving tasks test whether agents can produce correct patches, and test-generation tasks check if agents can create regression tests that fail on buggy code and pass after the fix.

Results show that even strong AI agents and large models still perform poorly on DevOps tasks. The best agent achieved around 58% accuracy in build and configuration, but less than 25% in monitoring, issue fixing, and test generation. The study finds that current systems struggle to plan multiple steps, use DevOps tools properly, and reason about dynamic system states. It concludes that existing models are far from being able to automate the entire DevOps cycle.

**Strengths:**

The paper’s main strength lies in its clear motivation and solid contribution to an underexplored area. It identifies that while many existing benchmarks test AI on programming or debugging tasks, none assess performance across the complete DevOps pipeline, which involves building, monitoring, fixing, and testing software systems. By defining this broader scope, the authors push evaluation toward real-world DevOps scenarios that require understanding of environments, tool interaction, and sequential task execution. This is a meaningful extension beyond static code generation benchmarks and provides a realistic testbed for future research.

Another strong point is the benchmark’s detailed and careful construction. It covers nearly seven hundred real-world tasks from more than thirty Java and Go projects, including both real GitHub issues and synthetic cases designed by experts to represent frequent DevOps problems. Each task is clearly defined with inputs, expected outputs, and evaluation criteria. The inclusion of four DevOps stages—build/configuration, monitoring, issue resolving, and test generation—ensures comprehensive coverage. The benchmark also integrates command-line tools commonly used in DevOps such as Maven, Gradle, JUnit, and iostat, allowing agents to operate in realistic conditions instead of simulated text environments.

The authors use contamination checks to prevent overlap with training data, manually reconstruct real-world project environments, and validate that each task is reproducible. They also test multiple well-known agent frameworks and models, providing a broad and transparent performance comparison. Their analysis goes beyond simple accuracy reporting; they identify specific failure patterns such as agents’ difficulty with tool usage, temporal reasoning, and multi-step planning. This combination of realism, technical rigor, and analytical depth makes the benchmark a valuable and trustworthy foundation for studying AI in software DevOps automation.

**Weaknesses:**

The paper, while ambitious and valuable, has several weaknesses that limit its overall contribution.

First, the scope is too broad relative to the experimental depth. Although it claims to evaluate the entire DevOps cycle, the number of tasks for some categories—such as monitoring or build/configuration—is relatively small compared to the hundreds of issue-resolving and test-generation cases. This uneven distribution makes the evaluation appear unbalanced, and it is unclear whether the benchmark fully captures the complexity of real-world DevOps workflows. Some parts, like deployment or CI/CD automation, are also missing, which weakens the claim of “end-to-end” coverage.

Second, the dataset construction process is heavily manual and lacks scalability. The paper mentions that each task can take over ten hours of expert work to reproduce and verify, suggesting that expanding the dataset or maintaining it over time will be difficult. The strong reliance on expert labor reduces the reproducibility and accessibility of the benchmark. In addition, while the authors describe contamination checks, they provide limited quantitative details or validation of how effective these checks are. Without transparency on contamination metrics or repository overlap, it’s hard to judge the fairness of the evaluation.

Third, the evaluation analysis is somewhat surface-level. While the results show that agents perform poorly, the discussion of why these failures occur remains descriptive rather than diagnostic. For example, the paper reports that agents fail in tool usage or temporal reasoning but does not analyze whether this is due to prompt design, insufficient model context, or poor environment instrumentation. There is also limited analysis of error types or task difficulty variations. The work lacks ablation or controlled experiments that could explain which aspects of the benchmark are truly challenging.

**Questions:**

1) The benchmark includes 698 tasks from 30+ Java and Go projects, but DevOps typically spans far more diverse environments and languages. How confident are the authors that these two languages and this limited set of projects represent the broader DevOps space, including cloud-native and containerized systems?
2) The paper claims to prevent contamination using prefix-completion tests, but no quantitative contamination rates are reported. Could the authors provide clearer evidence or benchmarks for how effective this process was across repositories?
3) The benchmark covers four stages but omits key aspects of DevOps such as deployment, CI/CD automation, rollback, and infrastructure management. Do the authors plan to extend to these in future work, or do they believe these tasks require fundamentally different agent capabilities?

---

> ### Author Response · Authors · 2025-11-22
> **Rebuttal by Authors (Part 1)**
>
> We thank the reviewer for the positive, insightful, and constructive feedback. Please refer to the following contents for our responses.
>
>
> ## Q1. Small task number for monitoring and build/configuration
>
> Monitoring and build/config tasks are inherently highly omplex in real-world systems, which means constructing these tasks, including injecting and reproducing the issues and design the automated judge, requires expert efforts and validation. For example, constructing a build task needs to understand the logics and code structure of the project, inject errors with the proper build file, rebuild the whole project to validate its correctness. As such, each issue carries high information density and diagnostic complexity. Constructing very large-scale tasks requires extensive efforts that way beyond one research project. As such, we keep the inital task number at a managable scale. Notably, DevOps-Gym is designed as an evolving benchmark. Our data creation pipeline is standardized and extensivable. We will actively maintain the project and adding new tasks, which also makes it a live benchmark. **During the rebuttal, we added four new monitoring tasks and six build tasks. These tasks also bring two new repos in Go and Java to our benchmark.** We will continue expanding these categories over time.  We have clarified this in the revised manuscirpt (Sec 5).
>
>
> ## Q2. Whether the selected languages and repos are representative?
>
> We believe our selected repos are representative for the following reasonings and will continue adding broader repos. First, we choose Java and Go because they jointly capture the core DevOps characteristics: large, compiled, dependency-heavy projects with complex build pipelines and long-running services. These features make them ideal for evaluating the end-to-end DevOps pipline. Besides, these languages are widely used in real-world projects that require DevOps. Second, the repos we selected are popular open-source ones, which contain the most common DevOps issues. Plus, we conducted decontamination filter. Third, we would like to point out that the challenges involved in current tasks are common DevOps issues that are generalizeable across languages. Besides, all tasks execute in fully containerized environments, which reflect cloud-native and service-oriented DevOps workflows without depending on external cloud providers. Moreover, our automated data collection pipeline is designed to accommodate new programming languages, making future language expansion straightforward.
>
> ## Q3. The dataset construction process is manual
>
> Our dataset construction is not fully manual. Large parts of the benchmark—including issue resolving, test generation, and a subset of build/configuration tasks—are collected through automated mining pipelines. Manual effort is required for tasks (e.g., monitoring metric, complex build task) that cannot be reliably reproduced automatically. These efforts are necessary and depend on the inherent complexity of the tasks, as these tasks require precise environmental reconstruction. Manual validation ensures high reproducibility and realistic coverage.
>
> ## Q4 Provide more details about data de-contamination
>
> We have updated Appendix B with additional clarification.
>
> Our contamination detection protocol employs a multi-metric approach to identify memorization. We sample 20 domain-specific code snippets per repository from CI/CD configurations, build files, and test files—artifacts less likely to appear in pre-training data. Using prefix completion with five complementary metrics (Levenshtein distance, consecutive token matches, mismatch position, longest common substring, and BLEU score), we classify snippets as high-risk when any metric exceeds predefined thresholds (e.g., 0.5 for BLEU score) indicating suspiciously accurate reproduction. The final contamination score for each repo is obtained by calculating the portion of high-risk snippets.
>
> The protocol effectively discriminates between contaminated and clean repositories. For example, beego and fastjson2 demonstrate 56% and 70% contamination for Claude-4-sonnet, while junit-framework shows only 10%.

---

> ### Author Response · Authors · 2025-11-22
> **Rebuttal by Authors (Part 2)**
>
> ## Q5. Add more steps in DevOps
>
> Our end goal is to cover the complete DevOps pipeline. However, due to the extensive effort needed, we select the four steps that can cover the core workflow of DevOps in this paper as the first work to assess Agents in DevOps. Note that, other stages CI/CD automation, or infrastructure management strongly depend on mutable external systems (cloud APIs, Kubernetes clusters, pipeline triggers, remote state), which requires way more efforts for task construction and automated evaluation (with necessary expert efforts to validate correctness). Besides, these steps primarily orchestrate build/monitor/resolve/test stages. As such, we start with the four steps that represent the core reasoning and tool-using capabilities required in DevOps while remaining feasible for rigorous, reproducible benchmarking.
> As a continuously evolve benchmark, we plan to expand our effort to more steps such as deployment and rollback. We have clarified this in the revised manuscirpt (Sec 5).
>
> ## Q6. Add more evaluation analysis
>
> We conducted detailed error analysis for build and monitor tasks and summarized common error types made by current agents. Details can be found in the revised manuscript (Sec 4.2)
>
> For Build/Config tasks, we diagnose three main error patterns: 1) Toolchain and environment instrumentation limitations (33%). Agents cannot validate configuration artifacts (e.g., unused imports, missing dependencies) because necessary validators are unavailable in the environment. 2) Multi-step reasoning failures (23\%). Agents resolve initial errors but lose track of remaining issues due to context preservation limitations and lack of iterative "fix-run-verify" loops. 3) Domain-specific knowledge gaps (37\%). These failures reflect technical complexity beyond current model capabilities, including build-system migrations, type compatibility issues, and platform-specific constraints.
>
> For monitoring, error analysis reveals four failure modes: 1) Inadequate monitoring methodology (37%): Agents use one-time commands instead of continuous observation, failing to track behavior over time. 2) Premature conclusion without monitoring (26%): Agents submit answers without executing any diagnostic commands. 3) Insufficient temporal granularity (11%): Agents use overly coarse sampling intervals (10-60s), missing transient anomalies like CPU spikes. 4) Interpretation failure (26%): Agents collect metrics correctly but fail to analyze them or draw incorrect conclusions.
> There are no false positives for monitor tasks. However, we observe that a substantial portion of seemingly correct predictions stem from the first error type: agents using inadequate one-time observations (e.g., a single top command) coincidentally yield correct answers on baseline instances with no injected errors. Since these instances represent normal system states, even a single snapshot can appear normal, leading agents to correctly conclude "no issues detected" despite employing improper monitoring methodology that would fail to detect actual anomalies.

---

> ### Author Response · Authors · 2025-11-26
> **Following up on our response**
>
> Dear Reviewer uo9F,
>
> Thank you for your valuable feedback on our manuscript. As the discussion period draws to a close, we wanted to follow up on our previous responses to your comments. We would greatly appreciate knowing whether our revisions and clarifications have adequately addressed your concerns, or if there are any remaining questions we can help resolve.
> We remain available to provide any additional information or clarification that would be helpful.
>
> Best,
>
> Authors

---

### Author Response · Authors · 2025-11-29
**Summary of response**

We sincerely thank all reviewers for their constructive and insightful feedback. Below, we summarize our key responses and updates to the paper:

## New Experiments and Analysis

> Statistical stability analysis (Reviewers EL6m, NdRZ)

We conducted five independent runs on a 50-task subset with Claude Code and OpenHands, showing consistent performance with low variance (STD < 2.5%) and modest improvement with Pass@5 aggregation. These results confirm the benchmark's stability and difficulty.

> Detailed error analysis (Reviewers uo9F, EL6m, jCaQ)

We added comprehensive error analysis for Build/Config and Monitoring tasks in Appendix D, categorizing failure modes and their frequencies. This provides deeper insights into current agent limitations.

> New baseline evaluation (Reviewer EL6m)

We evaluated Aider across all task categories, showing its lower performance compared to stronger frameworks like OpenHands and Claude Code.

> Dataset expansion (Reviewers uo9F, EL6m)

We added 10 new monitoring and build tasks during the rebuttal period, bringing two new repositories in Go and Java.

## Paper Updates and Clarifications

> Contamination detection (Reviewers EL6m, NdRZ)

We expanded Appendix B clarifying that the 20% threshold balances contamination control with dataset scale.

> Synthetic vs. real-world tasks (Reviewers EL6m, NdRZ)

We clarified the distribution (659 real-world, 39 synthetic) and justified synthetic tasks' necessity for monitoring (controlled anomaly patterns) and build implementation scenarios (rarely documented in GitHub issues).

> Ground truth and evaluation (Reviewers EL6m, jCaQ)

We emphasized that all tasks have execution-based, deterministic ground truth rather than subjective annotations. We also detailed the monitoring task evaluation procedure.

> Benchmark scope and limitations (Reviewers uo9F, NdRZ, jCaQ)

We clarified that the four selected stages represent the core reasoning and tool-using capabilities in DevOps while remaining feasible for rigorous benchmarking. Other stages like CI/CD primarily orchestrate these core stages and depend on mutable external systems that complicate evaluation. We plan to expand to deployment and rollback as future work.

---

### Meta-Review · Area_Chair_dQZy · 2026-01-06

**Summary:**

The reviews rate this submission with original ratings of 8, 6, 6, 4. This submission introduces DevOps-GYM. It is an e2e DevOps benchmark to measure the functions of build, configure, monitoring, issue resolving, tool use, and test case generation in containers.

The main concerns raised by the reviewers are the design method of the benchmark, including 1) coverage balance and representativeness (small monitoring/build counts; only Java and Go; missing the detailed CI/CD/deployment/infrastructure stages), 2) evaluation (single-run; missing statistical reporting; limited baselines), 3) synthetic-task validity and contamination transparency.

The authors' rebuttal strengthened this work by adding tasks, adding baselines, adding statistical stability checks, adding detailed error analysis, and clarifying contamination and synthetic-vs-real task composition. These make the benchmark more useful for the development of LLMs for DevOps. Therefore, in combination of the original average positive ratings and the rebuttal, this submission is recommended with an acceptance.

**Reviewer Concerns:**

Most concerns were light given the positive ratings, and addressed by the authors' rebuttal, including adding tasks, adding baselines, adding statistical stability checks, adding detailed error analysis, and clarifying contamination and synthetic-vs-real task composition.

The questions that are not fully addressed are difficult to face for most submissions, including the representativeness beyond Java and Go and cloud-native, the lack of CI/CD/deployment/rollback.

**Reviewer Scores:**

Reviewer EL6m and NdRZ: 6
Reviewer jCaQ: 8
Reviewer uo9F: 4, whose major concerns are on the lack of CI/CD/deployment/rollback, monitoring / build task counts, contamination. The authors' rebuttal managed to address most of them but not add CI/CD/deployment/rollback

---

### Decision · Program_Chairs · 2026-01-26

Accept (Poster)